# Machine Learning Model Based on Insulin Resistance Metagenes Underpins Genetic Basis of Type 2 Diabetes

**DOI:** 10.3390/biom13030432

**Published:** 2023-02-24

**Authors:** Aditya Saxena, Nitish Mathur, Pooja Pathak, Pradeep Tiwari, Sandeep Kumar Mathur

**Affiliations:** 1Department of Computer Engineering & Applications, Institute of Engineering & Technology, GLA University, Mathura 281406, India; 2Department of Medicine, Sawai Man Singh Medical College and Hospital, Jaipur 302004, India; 3Department of Endocrinology, Sawai Man Singh Medical College and Hospital, Jaipur 302004, India

**Keywords:** insulin resistance (IR), type 2 diabetes (T2D), machine learning, GSEA, artificial neural network, HOMA-IR, HOMA-B, GWAS

## Abstract

Insulin resistance (IR) is considered the precursor and the key pathophysiological mechanism of type 2 diabetes (T2D) and metabolic syndrome (MetS). However, the pathways that IR shares with T2D are not clearly understood. Meta-analysis of multiple DNA microarray datasets could provide a robust set of metagenes identified across multiple studies. These metagenes would likely include a subset of genes (key metagenes) shared by both IR and T2D, and possibly responsible for the transition between them. In this study, we attempted to find these key metagenes using a feature selection method, LASSO, and then used the expression profiles of these genes to train five machine learning models: LASSO, SVM, XGBoost, Random Forest, and ANN. Among them, ANN performed well, with an area under the curve (AUC) > 95%. It also demonstrated fairly good performance in differentiating diabetics from normal glucose tolerant (NGT) persons in the test dataset, with 73% accuracy across 64 human adipose tissue samples. Furthermore, these core metagenes were also enriched in diabetes-associated terms and were found in previous genome-wide association studies of T2D and its associated glycemic traits HOMA-IR and HOMA-B. Therefore, this metagenome deserves further investigation with regard to the cardinal molecular pathological defects/pathways underlying both IR and T2D.

## 1. Introduction

The global prevalence of insulin resistance (IR) has been estimated to range from 15.5 to 46.5% among adults [1]. It has been identified as a central pathophysiological factor of several endocrine–metabolic disorders, such as type 2 diabetes (T2D), high blood pressure, dyslipidemias, polycystic ovary syndrome, metabolic syndrome (MetS), cardiovascular diseases, etc. More significantly, it precedes and could lead to T2D, which now affects over 422 million people globally and accounts for 1.5 million deaths annually [2].

T2D is the most frequent subtype of diabetes and it is characterized by alterations of blood glucose levels due to varying combinations of IR and a relative deficiency of insulin secretion by the pancreatic β cells. If untreated, T2D may escalate to various microvascular and macrovascular complications such as retinopathy, diabetic kidney disease, peripheral neuropathy, atherosclerotic vascular disease, etc. Its underlying pathogenesis is complex and includes an approximately 40–70% contribution of genetic factors [3,4,5]. The majority of the genetic risk loci, however, have been found to decrease insulin secretion rather than its action [6]. 

IR is considered to be the major underlying pathophysiological defect in both obesity and early-stage T2D. It is also the major mechanistic link between them. This pathway is supposed to be driven by “adiposopathy” or “sick fat”, characterized by adipocyte hypertrophy and infiltration of pro-inflammatory cells (such as M1 macrophages and dendritic cells, etc.) into the expanding adipose tissue. The resulting heightened free-fatty-acid release, pro-inflammatory cytokine flux within the pathological adipose tissue, and ectopic fat deposition set the stage for heightened IR and lipotoxicity-mediated impairment in pancreatic β cells, together leading to overt T2D.

However, healthy obese individuals without complications of IR or T2D are frequently observed, indicating that the pathophysiology of IR and its transition to T2D is much more complicated than was earlier thought. Furthermore, the monogenetic disorder lipodystrophy, which is characterized by high IR status despite the substantial absence of adipose tissue, makes the pathophysiology of IR more elusive. At the molecular level, there might be some core set(s) of genes and gene-driven pathways that are shared by both IR and diabetes. These genes and pathways might also be responsible for the transition of IR to T2D. Therefore, identification of these genes would enable us not only to devise further clinical strategies to mitigate these diseases, but also to predict the transition from IR to T2D.

High-throughput gene expression methods such as DNA microarray and RNA-seq have proven useful in deciphering the pathophysiological processes occurring at the molecular level in various disease states. A huge number of gene expression datasets are now available in public databases like the Gene Expression Omnibus (GEO) and Array Express, which are routinely mined for the extraction of biological insights to further fundamental and clinical understandings. Problems associated with these studies include their low sample sizes due to ethical and other constraints, and the simultaneous measurements of tens of thousands of genes in these experiments. This high-dimensional nature makes the findings of these studies difficult to generalize to other related studies, which is a prerequisite for their clinical utility. One solution to this problem is the statistical meta-analysis of multiple gene expression datasets across related studies, which provides a robust set of metagenes that could be more confidently exploited for diagnostic and therapeutic purposes [7].

Various microarray meta-analysis studies have been conducted to decipher the molecular pathophysiology of IR and T2D. For example, Jung et al. (2018) conducted a meta-analysis of seven IR microarray studies and obtained drug signatures for two antidiabetic medicines, metformin and thiazolidinediones [8]. They also validated their signatures through cross-species analysis. Saxena et al. (2017–18) conducted combined system-level meta-analysis of IR and T2D datasets and highlighted adiposopathy—the inflammation of adipose tissue—as a central mechanism that leads to T2D [9,10]. However, one problem with large scale meta-analyses is that they yield a long list of metagenes which are difficult to interpret biologically, and it is practically difficult to exploit them further for clinical intervention due to the pleiotropic nature of the majority of genes in a genome.

In recent years, due to the availability of high volumes of clinical and molecular data, machine learning (ML) methods (also including deep learning methods) have begun to gain wide acceptance in the development of data-driven solutions to biomedical problems. These methods provide various mathematical and statistical models, which are provided with labeled datasets for supervised learning to enable them to predict the label or class of new data instances. Additionally, they can also be utilized for feature selection by identifying features or parameters in training datasets, which can significantly contribute to the models’ predictions. Machine learning has been used to develop predictive models for T2D using clinical features [11,12,13,14]; however, its use to predict T2D from gene-level features is relatively limited. Furthermore, the development of machine learning models based on IR gene expression could allow prognostication of the onset of T2D, and could be considered a novel approach with prospective clinical utility in T2D treatment.

It this study, we first attempted to obtain a robust IR metagene signature through meta-analysis of multiple microarray gene expression studies comprising samples of NGT and IR individuals.

Gene expression datasets are high-dimensional in nature and contain expression measurements of tens of thousands of genes across a limited number of samples. ML models based on these datasets may suffer problems with overfitting, which limits their predictive capability. To alleviate this problem, we first reduced the number of metagenes using an ML method—Least Absolute Shrinkage and Selection Operator (LASSO)—and obtained a core set of metagenes, hereafter termed key metagenes. These key metagenes were subsequently used to develop five baseline machine learning models based on LASSO, Support Vector Machine (SVM), eXtream Gradient Boosting (XGBoost), Random Forest (RF), and Artificial Neural Network (ANN).

We explored the potential of metagenes and key metagenes to explain the causation of T2D using GSEA-based enrichment analysis and disease-domain-specific tools, respectively. We also checked whether these key metagenes might represent the genetic basis of T2D, based on their presence in T2D and other glycemic traits identified in genome-wide association studies (GWASs). In addition to this direct evidence supporting the robustness of our ML models, we also checked for the presence of upstream transcription factors of these key metagenes in the GWASs as indirect evidence for the involvement of these genes in the genetic basis of T2D.

Our best-performing ML model demonstrated high prediction power in both cross-validation and test datasets. To sum up, the findings of present study endorse the use of our ML model in predicting the occurrence of T2D in IR individuals on the basis of the transcription signature of key metagenes. This ML model deserves further investigation of potential applications of these key metagenes in the prediction of transition from NGT to T2D in IR individuals, and of potential drug targets for the prevention and treatment of T2D.

## 2. Materials and Methods

### 2.1. Selection of Microarray Datasets and Meta-Analysis

The GEO database at the National Centre for Biotechnology Information was searched for gene expression studies including NGT and IR human tissue samples. As various insulin-responsive organs and tissues are involved in IR development, including pancreas, liver, skeletal muscle, kidneys, brain, small intestine, adipose tissue (subcutaneous and visceral), and peripheral blood mononuclear cells (PBMCs), we selected a total of nine gene expression datasets which profiled gene expression levels in these tissues (Table 1).

The GEOquery package [15] from R Bioconductor was used to download series matrix files for each selected study. Meta-analysis was carried out using the web-based tool Network Analyst [16]. To adjust for the batch effect among datasets, the ComBat function in the SVA R package was used. To derive meta-signatures, Fisher’s method was used on gene-level log-transformed *p*-values after adjustment of batch size. Metagenes were selected based on *p*-values < 0.05.

### 2.2. Enrichment Analysis of Metagenes

Computational validation of IR metagenes was conducted using gene set enrichment analysis [17] methods to assess whether these genes were enriched in phenotype-relevant biological processes. The GSEAPreranked method in GSEA allows the submission of metagenes ranked by their Z-scores. Enrichment analysis of metagenes was conducted using the Gene Ontology (GO) and Kyoto Encyclopedia of Genes and Genomes (KEGG) collections of the Molecular Signatures Database (MSigDB) [18] to biologically validate the obtained metasignature; the results of the enrichment analyses were later visualized using EnrichmentMap [19] in the Cytoscape [20] environment.

### 2.3. Selection of Key Metagenes and Their Biological Validation for T2D

The batch-effect-adjusted combined expression matrix was then filtered against the metagenes to obtain the IR meta-expression matrix. This matrix was subsequently used as the input for the ML method LASSO, which also performs feature selection and has been widely used in various biomedical studies to reduce numbers of genes [21,22,23,24]. The reduced set of metagenes obtained from LASSO was termed the “key metagenes”. As GO terms are not disease-specific, in order to explore the implications of these key metagenes in the disease domain, we used two tools—GLAD4U (Gene List Automatically Derived For You) [25] and DisGeNET [26]—to compile up-to-date gene–disease association information using text mining.

### 2.4. GWAS Evidence for Involvement of Key Metagenes in T2D

In order to further validate the roles of the 72 identified genes in diabetes, we checked the Type 2 Diabetes Knowledge Portal (https://t2d.hugeamp.org/; accesed on 23 November 2022) which houses genes that have been found to be associated with type 2 diabetes and other glycemic traits such as HOMA-IR and HOMA-B across multiple GWASs. The *p*-values for associations with phenotypes were calculated using the MAGMA (Multi-marker Analysis of GenoMic Annotation) method. In addition to finding GWAS-based direct associations of key metagenes, we also obtained a list of upstream regulators (transcription factors) of these genes using the Expression-to-Kinase (X2K) [27] webserver, which infers upstream regulatory networks from the signatures of differentially expressed genes through a combination of transcription factor enrichment analysis, protein–protein interaction network expansion, and kinase enrichment analysis. 

### 2.5. Construction of Machine Learning Models and Their Performance Evaluation 

The IR meta-expression matrix was further filtered for the key metagenes and the resulting matrix was then used to train five baseline ML models. SVM has been used to classify gene expression data for many years [28,29]; however, its use as a predictive model in the bioinformatics community has also begun to gain momentum in recent times [30]. XGBoost and RF are ensemble-based learning methods that assimilate multiple tree models to build a robust learner model. These methods have been used for the prediction and classification of gene expression data [31,32]. Deep learning models, particularly ANN, have also been used extensively in bioinformatics [33,34,35,36,37,38,39]. Details of the ML methods and their tuning parameters are presented in the Appendix A. 

The GEO dataset GSE64577, comprising 64 human adipose tissue samples, was used as a test dataset to check the performance of the ML models. These samples were drawn from a Mexican-American population under the Veterans Administration Genetic Epidemiology Study (VAGES) [40] and included 38 T2D and 26 NGT individuals as per their fasting plasma glucose level (T2D > 100 mg/dL). 

We used five-fold cross-validation and the performance of the ML models was evaluated using the following parameters: accuracy (A), precession (*p*), recall (r), sensitivity (Sen), specificity (Spe), F1 score, and Matthews correlation coefficient (MCC) from the confusion matrix for both the validation and the test dataset. These values were calculated using the indicators TP (true positive), FP (false positive), FN (false negative), and TN (true negative).
$$A=\frac{TP+TN}{TP+TN+FP+FN}$$
$$p=\frac{TP}{TP+FP}$$
$$r=\frac{TP}{TP+FN}$$
$$F1=\frac{2\times p\times r}{p+r}$$
$$MCC=\frac{TP*TN-FP*FN}{\sqrt{\left(TP+FP\right)\left(TP+TN\right)\left(TN+FP\right)\left(TN+FN\right)}}$$
where TP is the number of IR/T2D individuals predicted to be IR/T2D by the model; FN is the number of IR/T2D individuals incorrectly predicted to be NGT by the model; TN is the number of NGT individuals predicted to be NGT by the model; and FP is the number of NGT individuals incorrectly predicted to be IR/T2D by the model.

## 3. Results

NetworkAnalyst identified an IR metasignature of 2574 genes (*p* < 0.01), containing 1488 upregulated (Z-score > 0) and 1086 downregulated (Z-score < 0) genes. The derived IR meta-expression matrix for these metagenes had 367 expression datasets consisting of 182 NGT and 185 IR samples. 

GSEA-based functional analysis of the IR datasets showed enrichment in various GO terms such as “RNA metabolism”, “chromosome”, “transmembrane transport”, and “ubiquitin protein activity” (Figure 1a). The majority of these processes were found to be downregulated, likely due to weak insulin signaling in peripheral tissues in the setting of insulin resistance. Enriched KEGG pathways were “oxidative phosphorylation”, “type 2 diabetes”, and “ribosome” (Figure 1b). Enrichment of these pathways therefore provides biological validation of the IR signature. 

A total of 72 key metagenes were obtained from the meta-signature through LASSO-based feature selection. GLAD4U analysis showed robust enrichment of various diabetic terms (Figure 2) such as “diabetes mellitus”, “diabetes mellitus type 2”, “diabetes mellitus type 2 and obesity”, “diabetic ketoacidosis”, etc. (marked red). Various cardiovascular complication terms were also found, such as “ventricular premature complexes”, “shortened QT interval”, “ventricular fibrillation”, etc. (marked blue). Traditionally, diabetes mellitus is considered to be a disorder of glucose metabolism, whereas the roles of the disorders of lipid metabolism and adipose tissue dysfunction in the pathogenesis of atherosclerotic vascular disease are well known. However, this classical picture is changing rapidly due to appreciation of the fact that IR driven by adipose tissue dysfunction induces beta-cell dysfunction both directly via its exhaustion and indirectly via lipotoxicity. Therefore, adipose tissue also plays a major role in the pathogenesis of T2D. 

Additionally, cardiovascular diseases and phenotypes and terms related to renal and hepatic complications such as kidney diseases, fatty liver, carcinoma hepatocellular, urogenital neoplasm, and urination disorder were also found to be enriched. Six of the key metagenes—*INSR*, *MAP3K5*, *NDUFB8*, *NDUFS1*, *SDHB*, and *UQCRC2*—are involved in nonalcoholic fatty liver disease [41] (hsa04932), which is caused by a defect in insulin suppression of free fatty acid (FAA) disposal due to the induction of insulin resistance. Furthermore, a reduction in nitric oxide production due to IR has been reported to inhibit bladder smooth muscle cell growth and consequently lead to urination disorder [42]. 

DisGeNET analysis highlighted that 27 of the IR key metagenes (~37%)—*ALDH6A1, APOB, ARID5B, ATP2B2, ATXN1, B2M, CAT, CFB, CHI3L1, COL8A1, EIF2AK2, FHL2, GATM, HDAC4, HIPK3, HOMER1, IGFBP5, INSR, MAP3K5, MMP9, NDUFS1, RASSF7, REV3L, SLC19A2, UROD, WAS*, and *ZEB1*—were found to be associated with T2D or its related variants such as alloxan, autoimmune, monogenic, neonatal, ketosis-prone, sudden-onset, insipidus, fibrocalculous pancreatic, phosphate, gestational, post-transplant, lipoatrophic, streptozotocin, brittle, and maturity-onset diabetes, among others (Figure 3). 

To assess the pathway-based functional assignment for these genes, we selected the human collection of WikiPathways and found enrichment of the following 15 pathways (*p* < 0.05), including “insulin signaling”, which has a conspicuous role in the IR phenotype (Table 2).

The pathway “FTO obesity variant mechanism” is relevant to the T2D phenotype as SNPs in the Fat mass and Obesity-associated (FTO) gene have been found to be associated with adiposity and risk of obesity in multiple populations [43]. Due to involvement of FTO in energy homeostasis, it could link IR to T2D. Disturbance in the “folate metabolism” pathway has been reported to induce glucose and lipid metabolism disorders in animal studies [44], suggesting its involvement in IR. The pathway “selenium micronutrient network” also influences adipocyte physiology and modifies the risk of developing T2D. In one study, high Se and hs-CRP concentrations were found to be associated with high HbA1c levels in various BMI groups [45]. Another pathway, “vitamin B12 metabolism”, is also relevant as high prevalence of low B12 levels have been shown in European (27%) and South Indian (32%) patients with type 2 diabetes (T2D) [46]. Angiopoietin like 8 (*ANGPTL8*) is involved in the regulation of lipid metabolic processes and triglyceride homeostasis, so the term “Angiopoietin Like Protein 8 regulatory pathway” is also relevant to IR pathophysiology. AGE/RAGE signaling increases oxidative stress to promote diabetes-mediated vascular calcification through activation of Nox-1 and decreased expression of SOD-1. Different pharmacological interventions are underway to regulate the AGE/RAGE pathway to decrease the severity of this diabetic complication [47]. Enriched pathways therefore showed the biological validity of the key metagenes. 

Key metagenes were further checked against T2D GWASs to ascertain their implications in the genetic architecture of the disease. Seventeen genes were mapped to genetic loci associated with T2D in GWASs. These genes were *CFB, PRSS3, ARID5B, REV3L, CROCC, HOMER1, NDUFB8, DDX17, DCAF8, EIF2AK2, ZEB1, ATF6B, UROD, MAP3K5, MMP9, STK24,* and *WNT4*; three genes with HOMA-IR-related loci, *MATK, FHL2,* and *PHKB;* and two with HOMA-B-related loci, *HS3ST1* and *ALDH6A1*. The gene *INSR* has been found to show association with all the three traits.

The human genome consists of 24,500 protein-coding genes, of which 4049 have been identified to have T2D GWAS signals (*p* < 0.05), so the probability of picking a T2D gene by chance would be 0.17 (i.e., 4049/24,500). 

To check the statistical robustness of our key metagenes, we estimated the probability of finding 17 T2D genes by chance across 72 key metagenes using a binomial test (n = 72, x = 17, *p* = 0.17) and obtained a low probability (*p* = 0.03), and thus concluded that our key metagene set was nonrandomly enriched with T2D genes.

We carried out further analysis by mapping of the regulatory elements of the key metagenome using the X2K webserver. The webserver analyzed the key metagenome of IR and reported 83 transcription factors (including 2 key metagenes). Interestingly, 42 of these transcription factors were found to be associated with T2D and its related traits in GWASs. In other words, we could provide in silico evidence of the mapping of key metagenome genes to T2D and related loci identified in GWASs and shed light on their potential role in the genome-to-phenome trajectory of T2D. Therefore, our ML model based on expression profiles of these 72 key metagenes could be considered highly robust for predicting diabetes. 

All five machine learning models demonstrated good predictive capabilities upon five-fold cross-validation. Detailed information of the various evaluation indicators is shown in Table 3.

The classification accuracy was visualized using receiver operator characteristic (ROC) plots for all the models and all showed an area under the curve (AUC) > 75%, with ANN returning the maximum AUC of 95% (Figure 4). 

The best-performing model, ANN, was subsequently evaluated for its performance on the test dataset. Out of 64 samples (38 T2D: 26 NGT), it predicted 28 samples as true positive, 21 samples as true negative, 10 samples as false negative, and 5 samples as false positive, and showed 73% sensitivity, 80% specificity, and an overall F1 score of 0.78 (overall accuracy 76%). Despite the high biological noise in the gene expression data, differences among microarray platforms, and also marked differences in the molecular pathologies of IR and T2D due to chronic hyperglycemia in the latter condition, our IR-based ML model achieved fairly good accuracy in differentiating individuals with T2D from non-T2D people. 

## 4. Discussion

In this study, we developed a machine learning model that could differentiate an individual with T2D from a NGT individual on the basis of expression of a key metagenome of IR in insulin-responsive tissues, particularly adipose tissue, with almost 70 to 75 percent accuracy in a cross-sectional study. This key metagenome was created via the meta-analysis of publicly available databases of transcription profiles of various insulin-responsive tissues obtained from IR non-T2D individuals with diverse ethnic backgrounds. The genes in this key metagenome enrich several known pathways of not only diabetes mellitus, but also cardiovascular disease, specifically cardiac arrhythmias, kidney diseases, fatty liver, carcinoma hepatocellular, urogenital neoplasm, urination disorder, etc. The genetic loci regulating the expressions of most of these genes are mapped to genomic regions identified in GWASs as associated with diabetes and related traits. Therefore, these findings suggest that the key metagenome of insulin resistance (IR) also plays a functional role in the genetic susceptibility to T2D via the enrichment of several known pathophysiological molecular pathways of T2D in metabolically active tissues. As the IR key metagenome can be mapped to GWAS-identified genetic loci of diabetes, this metagenome is unlikely to be merely an acquired pathophysiological adaptation to impaired insulin signal transduction; rather, it is a primary genetic defect. 

There are several other implications of these findings. For example, T2D and IR are complex traits and their genetic susceptibility is generally believed to be omnigenic in nature. However, in this study, out of almost 30,000 genes in the human genome, only 72 genes were found to be of functional significance at the transcriptome level to the manifestation of genetic susceptibility to T2D via the known molecular pathways of IR. However, a limitation of the present study is that the source of this cross-sectional meta-analysis was data from publicly available depositories, and the mapping and functional roles were imputed in silico. Though we have estimated that our metagenes are associated with T2D GWAS signals, due to the nature of our study being cross-sectional, these gene expression results might not be free from reverse causation. Therefore, there is a need for validation of the role of this metagenome through deciphering of the genome-to-phenome trajectory of diabetes using replication studies in different insulin-responsive tissues. 

If this metagenome proves to be a gold standard transcriptomic signature of IR in diabetes, then it would also deserve further inquiry regarding clinical applications like diagnostic molecular markers and drug target discovery. However, the most important and challenging question is as follows: How should these metagenes obtained from insulin-responsive tissues in research settings be measured/assessed in clinical practice? Therefore, from the clinical point of view, there is a need to identify not only the clinical, biochemical, and radiological parameters that show associations with the key metagenome, but also the gene sequence polymorphisms and the circulatory biomarkers associated with this metagenome. 

IR and beta-cell dysfunction are two major pathophysiological defects in T2D. IR has traditionally been considered to be caused by environmental factors, whereas the genetic susceptibility to develop T2D possibly plays a major role in beta-cell failure. However, identification of the key metagenome of IR, its imputed mapping to T2D GWAS loci, and functional enrichment of known T2D pathways and validation in adipose tissues of T2D individuals suggests a role of genetic factors in driving IR. Additionally, this metagenome is derived from meta-analysis of several insulin-responsive tissue transcription profiles. Therefore, it points towards a common molecular thread shared by most metabolically active insulin-responsive tissues and it could serve as “the cardinal molecular pathology” of IR, T2D, and MetS. However, this concept deserves further investigation. 

Another important finding of this study is that the genes in the key metagenome of IR enrich both diabetes- and cardiac-arrhythmia-related pathways. Therefore, they shed light on a common functional genomic defect underlying these two diseases. The relationship between diabetes and cardiac arrhythmias is complex and it is not yet fully understood. However, relationships between diabetes and arrhythmias have been reported [48]. There are several potential mechanisms of arrhythmia in diabetes, such as increased blood glucose levels, glucose fluctuation, hypoglycemia, autonomic dysfunction, alterations in the architecture of the heart including fibrosis, fat deposition, hypertrophy, etc. In addition, the finding of a key metagenome linking IR with cardiac arrhythmias in the present study points towards another novel molecular mechanism shared by these two diseases. 

## 5. Conclusions

In conclusion, a machine learning model trained with the key metagenome of IR can differentiate individuals with T2D from NGT with moderate accuracy on the basis of transcription profiles of adipose and other insulin-responsive tissues. The mapping of this metagenome to GWASs identified loci of diabetes, and the enrichment of known molecular pathways of diabetes suggest a primary role of this metagenome in the pathogenesis of diabetes rather than merely a pathophysiological response to impaired insulin signaling. Therefore, this metagenome deserves further investigation in terms of the cardinal molecular pathological defects/pathways underlying both IR and diabetes. 

## Figures and Tables

**Figure 1 biomolecules-13-00432-f001:**
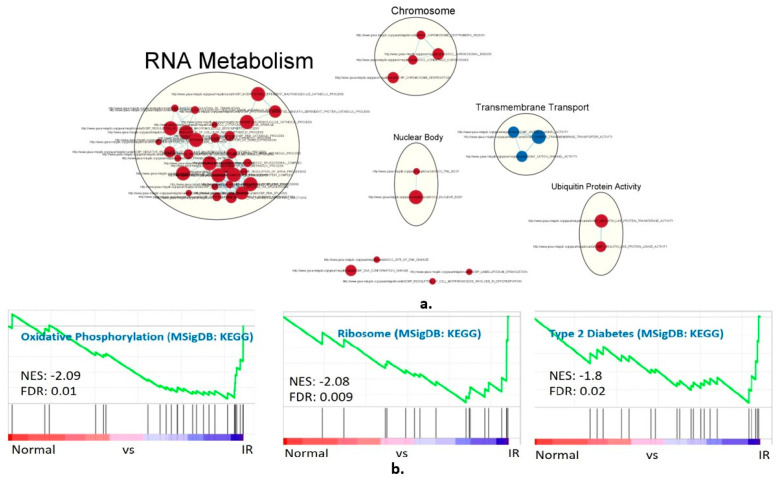
Enrichment analysis of (**a**) IR metagenes using Z-scores of the meta-analysis and Gene Ontology (GO) gene sets. (**b**) GSEA analysis of IR metagenes using KEGG in the Molecular Signatures Database (MSigDB). NES, normalized enrichment score; FDR, false discovery rate.

**Figure 2 biomolecules-13-00432-f002:**
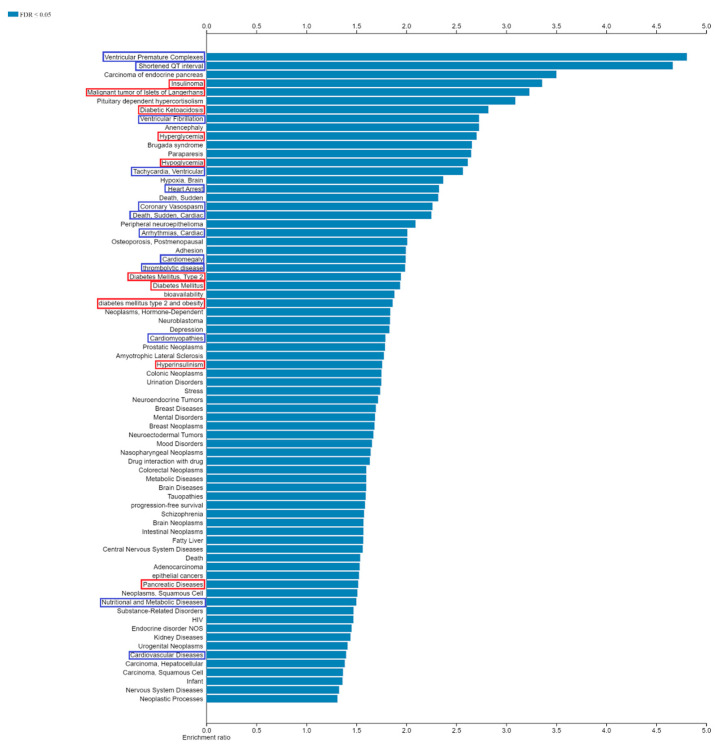
Enrichment of disease terms for IR key metagenes using GLAD4U with false discovery rate <0.05 and number of genes in a category >4.

**Figure 3 biomolecules-13-00432-f003:**
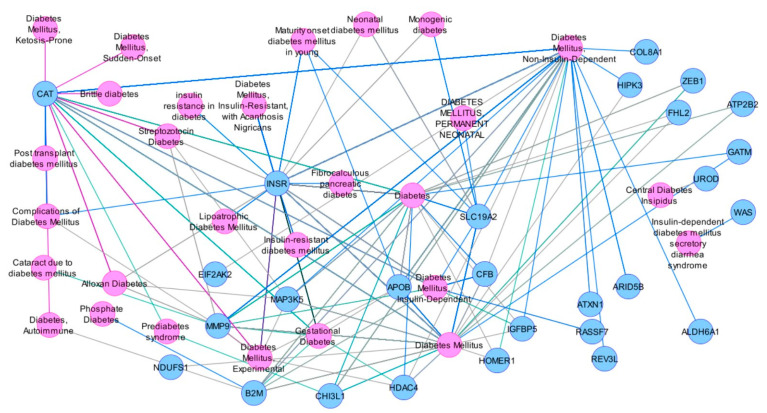
Disease–gene network for IR key metagenes for type 2 diabetes and related disorders, created using DisGeNET.

**Figure 4 biomolecules-13-00432-f004:**
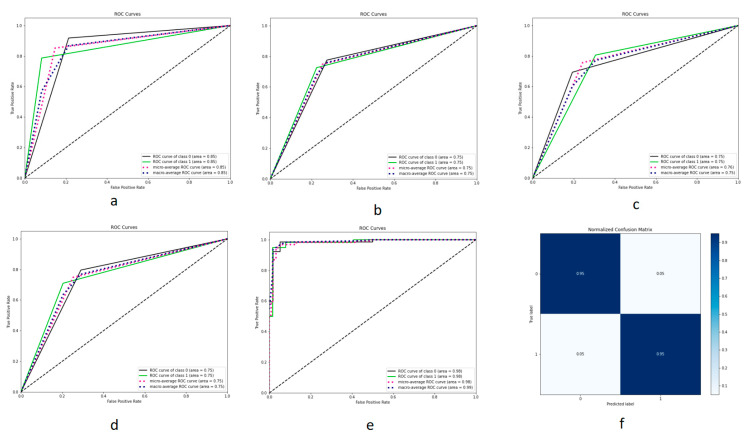
Receiver operator characteristic curves for (**a**) Least Absolute Shrinkage and Selection Operator (LASSO), (**b**) Random Forest (RF), (**c**) Support Vector Machine (SVM), (**d**) eXtream Gradient Boosting (XGBoost), and (**e**) Artificial Neural Network (ANN), and (**f**) confusion matrix for best-performing ANN model.

**Table 1 biomolecules-13-00432-t001:** Microarray samples used in meta-analysis for IR.

S. No	GEO Series	Tissue	Place of Study	Disease Phenotype	
				Insulin Resistance (IR)	Normal Glucose Tolerance (NGT)
1	GSE6798	Skeletal muscle	Department of Hematology in Roskilde Hospital, Roskilde, Denmark.	IR = 16	13
2	GSE15773	Subcutaneous adipose tissue (SAT) and visceral adipose tissue (VAT)	Department of Molecular Medicine at University of Massachusetts, Worcester, MA, USA	IR = 4 (SAT) IR = 5 (VAT)	5 (SAT) 5 (VAT)
3	GSE20950	Subcutaneous adipose tissue (SAT) and visceral adipose tissue (VAT)	Department of Molecular Medicine at University of Massachusetts, Worcester, MA, USA.	IR = 9 (SAT) IR = 10 (VAT)	10 (SAT) 10 (VAT)
4	GSE22309	Skeletal muscle	Department of Biostatistics at University of Alabama, Birmingham, AL, USA	IR = 20	20
6	GSE26637	Subcutaneous adipose tissue	Department of Institute for Molecular Medicine Finland (FIMM), University of Helsinki, Helsinki, Finland.	IR = 5 (Fasting) IR = 5 (Hyperinsulinemia)	5 (Fasting) 5 (Hyperinsulinemia)
7	GSE34526	Granulosa cells	Department of Zoology at University of Delhi, Delhi, India.	IR = 16 (*PCOS*)	12 (*PCOS*)
8	GSE36297	Vastus lateralis muscle	Department of Hematology, Roskilde Hospital, Roskilde, Denmark.	IR = 6	10
9	GSE64567	Fasted subcutaneous abdominal adipose tissue (FAT)	Department of Medicine at University of Texas, Health Sciences Center at San Antonio, San Antonio, TX, USA.	IR = 38	26

**Table 2 biomolecules-13-00432-t002:** List of pathways enriched by the IR key metagenes against the human collection of WikiPathways with *p* < 0.05.

S. No.	Enrichment FDR	Genes in List	Total Genes	Functional Category
1	4.4E-03	2	8	FTO Obesity Variant Mechanism
2	4.4E-03	4	103	Electron Transport Chain
3	1.1E-02	3	66	Folate Metabolism
4	1.7E-02	3	85	Selenium Micronutrient Network
5	2.8E-02	3	110	DNA Damage Response (only ATM dependent)
6	2.9E-02	2	38	Amyotrophic lateral sclerosis (ALS)
7	3.0E-02	2	50	Vitamin B12 Metabolism
8	3.0E-02	3	132	Angiopoietin Like Protein 8 Regulatory Pathway
9	3.0E-02	2	52	Translation Factors
10	3.0E-02	3	155	Myometrial Relaxation and Contraction Pathways
11	3.0E-02	3	160	**Insulin Signaling**
12	3.0E-02	2	45	ATM Signaling Network in Development and Disease
13	3.0E-02	3	159	Epithelial to mesenchymal transition in colorectal cancer
14	3.0E-02	2	60	Oxidative phosphorylation
15	3.1E-02	2	66	AGE/RAGE pathway

**Table 3 biomolecules-13-00432-t003:** The evaluation indicators of the five machine learning models.

Evaluation Indicators	LASSO	SVM	XGBoost	Random Forest	ANN
False Positive Rate | Type I error	0.22	0.20	0.08	0.30	0.04
False Negative Rate | Type II error	0.27	0.29	0.21	0.19	0.05
True Negative Rate | Specificity	0.77	0.79	0.91	0.69	0.95
Negative Predictive Value	0.69	0.68	0.81	0.73	0.95
False Discovery Rate	0.19	0.18	0.09	0.23	0.051
True Positive Rate | Recall | Sensitivity	0.72	0.70	0.78	0.80	0.94
Positive Predictive Value | Precision	0.80	0.81	0.90	0.76	0.94
Accuracy	0.74	0.74	0.85	0.75	0.95
F1 Score	0.76	0.75	0.84	0.78	0.94
Matthews Correlation Coefficient MCC	0.49	0.50	0.71	0.50	0.9
ROC AUC score	0.75	0.75	0.85	0.75	0.95

## Data Availability

Correspondence and requests for data and materials should be addressed to S.K.M.

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
