# Peer review of "Machine Learning Model Based on Insulin Resistance Metagenes Underpins Genetic Basis of Type 2 Diabetes"

_biomolecules, 2023, doi:10.3390/biom13030432_

Round 1

Reviewer 1 Report

The authors developed machine learning-based models by training on insulin resistance-associated gene expressions in insulin-responsive tissues to differentiate individuals with and without type 2 diabetes. These 72 so-called "key meta genes" yielded 73% accuracy in classifying individuals with type 2 diabetes in a test set of human adipose tissue samples. The authors deployed a wide selection of bioinformatics and machine-learning approaches to investigate their questions.

General comments:

-Abbreviations are sometimes not correctly used (e.g., NGT in line 18). The first occurrence needs to be spelled out; after that, only the abbreviations should be written. Genome-wide association Studies (GWAS) have been repeated in lines 106,156, and 276.

-Good to add test set information to the abstract; 64 human adipose tissue samples

-Avoid referring to individuals as insulin-resistant or diabetic patients; use individuals with insulin resistance or T2D instead.  

- Good to rewrite Lines 93-102 in explaining the need for the feature selection step; make it concise and avoid repeating the reasons. 

-Line 131, perhaps it was meant log-transformed gene level instead of log-transformed P-values!

-Details of ML methods and tuning parameters can be added to the supplementary file and are not necessary to be included in the main manuscript, specifically Lines 171-178 and Figure 1.

-Not sure whether the meta-analyses are conducted by considering the study's population. If not considered, this might cause bias, and therefore not possible to generalize the developed models. It might even decrease the prediction ability of the models in specific populations.

-Subsection 2.6 can be added to the previous (2.5) after briefing it.

-Line 191 correct to True Positives.

-List these selected 72 key meta genes in the Supplementary.

-Line 233 does not seem to have the right citation (42?)

 -Line 235: NO production ??

-An interesting way of testing the robustness of the key meta genes in lines 284-287. However, this can be more appropriately tested by randomly selecting hundreds of gene sets and comparing them with the key meta genes in association with T2D.

-Figure 5 is missing.

-Line 312: F1 should be corrected to perhaps 0.78

Author Response

Respected Reviewer,

Thank you very much for minutely reviewing our manuscript and imparting very useful suggestions and corrections. I have gone through all of them and herewith presenting a point-by-point response for your kind perusal.

Thanks

General comments:

Comment 1: Abbreviations are sometimes not correctly used (e.g., NGT in line 18). The first occurrence needs to be spelled out; after that, only the abbreviations should be written. Genome-wide association Studies (GWAS) have been repeated in lines 106,156, and 276.

Response: Corrected as suggested.

Comment 2: Good to add test set information to the abstract; 64 human adipose tissue samples

Response: added as suggested.

Comment 3: Avoid referring to individuals as insulin-resistant or diabetic patients; use individuals with insulin resistance or T2D instead.  

Response: Corrected as suggested.

Comment 4: Good to rewrite Lines 93-102 in explaining the need for the feature selection step; make it concise and avoid repeating the reasons. 

Response: rewritten as suggested.

Comment 5: Line 131, perhaps it was meant log-transformed gene level instead of log-transformed P-values!

Response: it is gene level log-transformed P-values with no comma between gene level and log-transformed P-values which has now been removed.

Comment 6: Details of ML methods and tuning parameters can be added to the supplementary file and are not necessary to be included in the main manuscript, specifically Lines 171-178 and Figure 1.

Response: Supplementary file has now been enclosed with manuscript as suggested.

Comment 7: Not sure whether the meta-analyses are conducted by considering the study's population. If not considered, this might cause bias, and therefore not possible to generalize the developed models. It might even decrease the prediction ability of the models in specific populations.

Response: we followed usual machine learning practise. Model were trained and 5-fold validated using IR datasets but tested on a separate dataset of 64 T2D and NGT individuals and therefore models offer generalizability.

Comment 8: Subsection 2.6 can be added to the previous (2.5) after briefing it.

Response: added as suggested.

Comment 9: Line 191 correct to True Positives.

Response: Corrected

Comment 10: List these selected 72 key meta genes in the Supplementary.

Response: Listed in Supplementary file as suggested. 

Comment 11: Line 233 does not seem to have the right citation (42?)

Response: It is correct as KEGG has instructed to cite their publication wherever their data source is mentioned.

Comment 12: Line 235: NO production??

Response: Full form of Nitric Oxide is now added in place of NO.

Comment 13: An interesting way of testing the robustness of the key meta genes in lines 284-287. However, this can be more appropriately tested by randomly selecting hundreds of gene sets and comparing them with the key meta genes in association with T2D.

Response: We believe our approach to be statistically robust and therefore allow us for the same. 

Comment 14: Figure 5 is missing.

Response: it has now been added. Thanks for acknowledging us this serious mistake. We have re-named all the images as Figure 1 has now moved to Supplementary File.

Comment 15: Line 312: F1 should be corrected to perhaps 0.78

Response: Corrected it was indeed 0.78. again thanks for acknowledging us this serious mistake.

Reviewer 2 Report

The manuscript entitled "Machine Learning Model based on Insulin Resistance Meta-genes underpins Genetic basis of Type 2 Diabetes" was clearly written by Aditya Saxena et al.,  The authors claimed that the pathways of Insulin resistance shared with Type 2 Diabetes mellitus were not clearly understood.  to understand the pathways they demonstrated through Machine Learning approaches.  They have employed five baseline ML models such as LASSO, SVM, XGBOOST, RANDOM FOREST, ANN.  

1.  The author should describe the outcome achieved from the each ML pattern.

2. Figure 5 cannot be identified in Page number 11.

3.  Explaining the results in the graphical representation could make a better presentation

Author Response

Respected Reviewer,

Thank you very much for minutely reviewing our manuscript and imparting very useful suggestions and corrections. I have gone through all of them and herewith presenting a point-by-point response for your kind perusal.

Thanks

General comments:

Comment 1: The author should describe the outcome achieved from each ML pattern.

Response: We have provided detailed information of various evaluation indicators in table 3. The classification accuracy of each ML pattern was also visualized using Receiver Operator Characteristics (ROC) and Area Under Curve (AUC) for each ML are also illustrated in recently added Figure 4 (earlier it should be Figure 5).

Comment 2: Figure 5 cannot be identified in Page number 11.

Response: It has now been added. Thanks for acknowledging us this serious mistake. We have re-named all the images as Figure 1 has now moved to Supplementary File.

 Comment 3: Explaining the results in the graphical representation could make a better presentation.

Response: A Graphical abstract has now been added

Round 2

Reviewer 1 Report

The authors have addressed the comments adequately.

Following comment 7, this is not about the machine learning approach and the usual practice. We are talking about meta-analyses, where you first select your input data through meta-analyses and end up with table 1. You have done your meta-analysis through NetworkAnalyst. This can basically be added to the discussion by mentioning that different populations have been considered for selecting these genes. Still, this sounds slightly biased, so it is good to acknowledge and inform whether it is only focused on data from European ancestry (e.g.) or is based on data from different populations. 

Comment 7: Not sure whether the meta-analyses are conducted by considering the study's population. If not considered, this might cause bias, and therefore not possible to generalize the developed models. It might even decrease the prediction ability of the models in specific populations.

Response: we followed usual machine learning practise. Model were trained and 5-fold validated using IR datasets but tested on a separate dataset of 64 T2D and NGT individuals and therefore models offer generalizability.

Reviewer 2 Report

Authors are tried to answer the reviewer questions.  It is commended.